# Fine mapping of the antigenic epitopes of the Gc protein of Guertu virus

**Meilipaiti Yusufu[1]☯, Ayipairi Abula[1]☯, Boyong Jiang[1], Jiayinaguli Zhumabai[1], Fei Deng[2], Yijie Li[1], Yujiang Zhang[3]\*, Juntao Ding[1]\*, Surong Sun[1]\***

**1** Xinjiang Key Laboratory of Biological Resources and Genetic Engineering, College of Life Science and Technology, Xinjiang University, Urumqi, China, **2** State Key Laboratory of Virology, Wuhan Institute of Virology, Chinese Academy of Sciences, Wuhan, China, **3** Center for Disease Control and Prevention of Xinjiang Uygur Autonomous Region, Urumqi, China

☯ These authors contributed equally to this work.

\* sr_sun2005@163.com (SS); dingjuntao2004@126.com (JD); xjsyzhang@163.com (YZ)

**Data Availability Statement:** All relevant data are within the manuscript.

**Funding:** This work was supported partly by grants from the National Natural Science Foundation of China (No. 81760365, 81960369 to S. R. S.). And

## Abstract

Guertu virus (GTV), a newly discovered member of the genus *Banyangvirus* in the family *Phenuiviridae*, poses a potential health threat to humans and animals. The viral glycoprotein (GP) binds to host cell receptors to induce a neutralizing immune response in the host. Therefore, identification of the B-cell epitopes (BCEs) in the immunodominant region of the GTV Gc protein is important for the elucidation of the virus–host cell interactions and the development of GTV epitope assays and vaccines. In this study, an improved overlapping biosynthetic peptide method and rabbit anti-GTV Gc polyclonal antibodies were used for fine mapping of the minimal motifs of linear BCEs of the GTV Gc protein. Thirteen BCE motifs were identified from eleven positive 16mer-peptides, namely EGc1 ($^{19}$KVCATT-GRA$^{27}$), EGc2 ($^{58}$KKINLKCKK$^{66}$), EGc3 ($^{68}$SSYYVPDA$^{75}$), EGc4 ($^{75}$ARSRCTSVRR$^{84}$), EGc5 ($^{79}$CTSVRRCRWA$^{88}$), EGc6 ($^{90}$DCQSGCPS$^{97}$), EGc7 ($^{96}$PSHFTSNS$^{103}$), EGc8 ($^{115}$AGLGFSG$^{121}$), EGc9 ($^{148}$ENPHGVI$^{154}$), EGc10 ($^{179}$KVFHPMS$^{185}$), EGc11 ($^{230}$QAGMGVVG$^{237}$), EGc12 ($^{303}$RSHDSQGKIS$^{312}$), and EGc13 ($^{430}$DIPRFV$^{435}$). Of these, 7 could be recognized by GTV IgG-positive sheep sera. Three-dimensional structural analysis revealed that all 13 BCEs were present on the surface of the Gc protein. Sequence alignment of the 13 BCEs against homologous proteins from 10 closely related strains of severe fever with thrombocytopenia syndrome virus from different geographical regions revealed that the amino acid sequences of EGc4, EGc5, EGc8, EGc11, and EGc12 were highly conserved, with 100% similarity. The remaining 8 epitopes (EGc1, EGc2, EGc3, EGc6, EGc7, EGc9, EGc10, and EGc13) showed high sequence similarity in the range of 71.43%–87.50%. These 13 BCEs of the GTV Gc protein provide a molecular foundation for future studies of the immunological properties of GTV glycoproteins and the development of GTV multi-epitope assays and vaccines.

## Introduction

Ticks, the species of which are either generalist or specialist parasites, are responsible for transmission of a variety of pathogens to animals and humans [1]. The incidence of tick-borne viral

the Science Research Key Project of Xinjiang
Education Department (No. XJEDU2019I002 to S.
R. S.). The funders had no role in study design,
data collection and analysis, decision to publish, or
preparation of the manuscript.

**Competing interests:** The authors have declared
that no competing interests exist.

diseases has increased over the past few decades, thereby compromising the health of humans
significantly. In recent years, tick-borne severe fever with thrombocytopenia syndrome virus
(SFTSV) and heartland virus (HRTV) infections have been shown to be associated with serious
human diseases and fatalities in the USA and East Asian countries [2–4]. Studies on SFTSV
and HRTV indicate that novel tick-borne viruses may pose a significant threat to public health.
Therefore, the discovery and investigation of novel tick-borne viruses are of great importance
for ensuring the control and prevention of potential disease outbreaks.

Guertu virus (GTV), which is a member of the newly defined genus *Banyangvirus* belong-
ing to the family *Phenuiviridae* of Order *Bunyavirales* [5], was first isolated in 2014 from *Der-
macentor nuttalli*, a tick found in the Guertu mountain region of Wusu, Xinjiang, China [6].
Researchers reported the morphological characteristics of GTV for the first time, and phyloge-
netic analysis showed that it was closely related to SFTSV [7–10]. The genome of GTV consists
of the following three single-stranded RNA segments: an S segment that contains open reading
frames that encode a nonstructural (NS) protein and a nucleoprotein (NP); an L segment that
encodes an RNA-dependent RNA polymerase; and M segment that encodes a glycoprotein
precursor that is cleaved into two mature envelope proteins (Gn and Gc) during co-transla-
tional modification [6]. Studies have shown that viral expression of the Gn protein is directly
correlated with the Gc protein-facilitated binding of the viral particles to the host cell surface
and that the Gc protein is efficiently translated from the Gn/Gc-encoding mRNA, thus driving
the fusion between the virus and the host cell membrane [11]. At present, presumable N-glyco-
sylation sites is located on SFTSV Gc glycoproteins [10, 11]. DC-SIGN has been identified as
the receptor factor on the host cell membrane [10]. Some studies have also shown that HS and
NMMHC-IIA may also be the entry factors related to cell entry of SFTSV [12]. Additionally,
Gn and Gc facilitate viral entry into the target cells and are the primary targets of neutralizing
antibodies [13].

B-cell epitopes (BCEs), which are recognized by B-cell receptors, induce cellular and
humoral immune responses in the host [14]. Therefore, the identification of viral BCEs would
deepen our understanding of the host immune response, provide candidate epitope peptides
for the development of epitope vaccines or the establishment of disease diagnostic methods,
and help to reveal the mechanisms of therapeutic antibodies [15, 16]. Specifically, the identifi-
cation of BCEs of the conserved domain or immunodominant region of the Gc protein of
GTV is crucial for the elucidation of the virus–host cell interactions and development of GTV
assays and vaccines [17]. However, up to date, there are no reports on epitope mapping of
GTV Gc segment and their fine localization.

The currently available epitope mapping methods include recombinant DNA techniques,
biosynthetic peptide (BSP) methods, chemically synthesized peptides or peptide microarrays,
and phage-display libraries [18–22]. The BSP method developed by Xu et al. in 2009 was used
for the identification of linear BCEs and minimal epitope motifs, and they successfully mapped
linear of BCEs of three structural proteins of human papillomavirus type 58, using anti-recom-
binant protein pAbs [23, 24]. In our previous study, this method was used to identify the linear
BCEs of structural proteins from a strain of Crimean-Congo hemorrhagic fever virus that was
isolated in Xinjiang, where its feasibility for epitope identification was demonstrated [25–27].
Additionally, Zhang et al. [17] used pAbs and the BSP method to identify nine BCEs on the
Gn protein of GTV. However, the identification of BCEs on the Gc protein of GTV has not
been reported thus far.

Therefore, this study was performed to identify the minimal BCEs on the Gc protein of
GTV, using biosynthetic peptide (BSP) methods [14], and rabbit anti-GTV-Gc pAbs. Subse-
quently, 13 minimal epitope motifs were obtained and their antigenicity was confirmed using
positive sera from sheep naturally infected with GTV. Python 2.7 was used to analyze the

three-dimensional (3D) conformation of the Gc protein, thus revealing that the identified BCEs were all located on the protein surface. This study provides fundamental data that can be used in future studies aimed at elucidating the immunological properties of the Gc protein of GTV and for developing diagnostic reagents and vaccines for infection prevention and control.

## Materials and methods

### Ethical statement

This study was approved by the Animal Ethics Committee of the Key Laboratory of Biore-sources and Genetic Engineering, Xinjiang University (Approval number: BRGE-AE001). The animal serum samples were collected using a random sampling method that did not involve sacrifice of the animals, and serum collection was performed according to this approved animal protocol [14].

### Antibodies

The rabbit pAbs raised against GTV Gc1 (aa 1–291) and Gc2 (aa 252–549) were prepared by the Wuhan Institute of Virology, Chinese Academy of Sciences. His-tag mouse monoclonal antibody, goat anti-mouse horseradish peroxidase-conjugated IgG (IgG-HRP), and goat anti-rabbit IgG-HRP were purchased from Beijing Solarbio Science & Technology Co., Ltd. The serum samples from sheep infected or not infected with GTV were previously identified by indirect immunofluorescence assay and reverse transcription-polymerase chain reaction (RT-PCR) assay [6].

### Construction of the truncated Gc fragments for recombinant plasmid expression

The prokaryotic expression plasmid pET-32a (+) containing the full-length Gc gene fragment of GTV strain DXM (GenBank accession no. KT328592.1) was stored at our laboratory. The prokaryotic expression vector pXXGST-3 was provided by Professor Xu Wanxiang of the Shanghai Institute for Biomedical Pharmaceutical Technologies, China. The Gc protein of GTV was divided into two truncated fragments Gc1 and Gc2 (Fig 1). These fragments were then PCR-amplified using the previously constructed pET-32a-GTV-Gc plasmid as a template

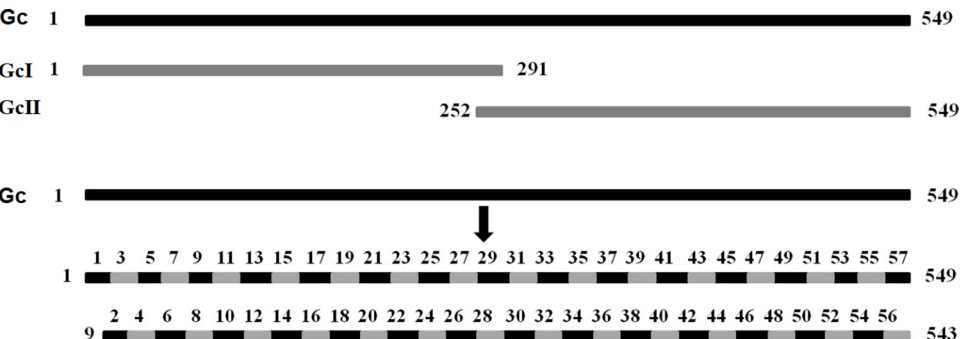

**Fig 1. Schematic of epitope mapping strategy.** (A) The black band indicates the full-length sequence of GTV-Gc, and the gray bands indicate two of truncated GcI and GcII segments. (B) Schematic of epitope mapping strategy involves 57 overlapping 16mer-peptides spanning Gc sequences.

[28] for the design of the following primers: GcI F: `CGGGATCCATGAAACTGGTTCGCTTAA CC`; GcI R: `AATGCGGCCGCTTATGATTTCAGTTCTTCAATTTCAC`; GcII F: `CGGGATCCATG AGTGGTGTTCCGACCCAG`; and GcII R: `AATGCGGCCGCTTAGGTCAGCAGCACGCCT`.

## Strategy for designing GTV Gc overlapping peptides

To map the linear BCEs of the GTV Gc protein, we designed 16mer-peptides with eight overlapping amino acid residues for the first round of antigenic peptide mapping (Fig 1). Additionally, a series of 8mer-peptides with seven overlapping amino acid residues that spanned every positive 16mer-peptides sequence were also designed for the second round of fine mapping of the BCE motif [29]. The GTV Gc protein was designed into fifty-seven 16mer-peptides using modified overlapping peptide biosynthesis. These 16mer-peptides were further shortened into a total of seventy-nine 8mer-peptides overlapping 7mer-peptides. The BSP method had the following features: (1) By selecting the truncated GST188 protein as the fusion tag for the recombinant expression of short peptides, the fragment size of the expressed fusion peptide was maintained within 21.5–22.5 kDa, which allowed its easy identification by sodium dodecyl sulfate–polyacrylamide gel electrophoresis (SDS-PAGE) and subsequent western blot analysis. This also facilitated the easy distinction of the short peptides from the proteins expressed by the host (*Escherichia coli*); (2) The antigenic proteins were first divided into 16mer-peptides with eight overlapping amino acid residues, following which basic and complete screening was performed for those containing BCEs; (3) Further truncation of the positive 16mer-peptides into 8mer-peptides with seven overlapping amino acid residues allowed the precise determination of the minimal epitope motif for each linear BCE [14].

## Construction and prokaryotic expression of the GTV Gc overlapping peptides

A suspension of bacterial cells harboring the empty prokaryotic expression plasmid pXXGST-3 was added to 10 mL of liquid medium containing 0.1% ampicillin and incubated overnight. The plasmid was then extracted using the Plasmid Mini-Prep Kit. The DNA fragments intended for incorporation into the plasmid were first digested with *BamH* I and *Sal* I at a constant temperature of 37°C for 4–5 h. According to the biosynthesis report of the biotechnology company, we considered the final concentration of each synthesized DNA fragment as 5 μM per tube. These fragments were mixed and heated to 95°C for 5 min in a constant temperature bath and then annealed by natural cooling to ambient temperature. The annealed oligonucleotides were mixed with the digested pXXGST-3 plasmid and then ligated at 16°C for 12–15 h. The vector was transfected into *E. coli* BL21(DE3) competent cells and cultured overnight, following which single clones were added to ampicillin-containing Luria broth and incubated overnight at 37°C under shaking conditions. On the next day, 1% of the overnight culture was inoculated into new Luria broth and cultured at 30°C under shaking conditions until the $OD_{600}$ reached 0.6–0.8. Heat induction at 42°C was then conducted for another 4 h [29].

## SDS-PAGE and western blot analysis of the GTV Gc overlapping peptides

The bacterial culture was centrifuged at 12000 rpm for 1 min, and the supernatant was discarded. After resuspending the cells in 160 μL of 1×phosphate-buffered saline, 40 μL of 5×loading buffer was added and the mixture was boiled at 100°C for 15 min and then centrifuged at 12000 rpm for 15 min. The protein samples were then separated by 15% SDS-PAGE. The plasmids expressing the correct protein size were send to Shanghai Sangon for sequencing. Western blot analysis was performed on the recombinant proteins, using rabbit anti-Gc pAbs to analyze their antigenicity. The proteins from strains with the correct sequencing

results were separated by 15% SDS-PAGE. Polyvinylidene difluoride (PVDF) membranes were purchased from Whatman International Limited. The separation gel was then removed with a transfer clamp and placed in an electrophoresis tank for electrotransfer of the proteins onto a PVDF membrane at 90 V for 90 min. Thereafter, the PVDF membrane was immersed in blocking buffer at 37˚C for 2 h, washed with Tris-buffered saline-Tween 20 (TBS-T), and then incubated with rabbit serum (1:300) at 4˚C overnight. Subsequently, the PVDF membrane was first rinsed with the washing buffer for 1 h and then incubated with HRP-labeled secondary antibody (1:2000) at 37˚C for 1 h. Finally, after rinsing the membrane with the washing buffer for 1 h, the ECL (Wuhan Boster Biological Technology Ltd.) chromogenic reagent was added and the membrane was imaged using the LAS-4000 chemiluminescence imaging system.

### Analyses of the similarities of the conformational antigenic epitopes and their localization on Gc by three-dimensional modeling

To analyze the similarity of each BCE with homologous proteins, Gc sequences of SFTSV strains from different countries and genetic lineages were downloaded from the GenBank database on the basis of the phylogenetic tree of SFTSV strains [9]. The BCE locations were experimentally identified from the 3D structure of the Gc protein using the PyMOL software (https://pymol.org/2/). Predictions of the secondary structure were conducted according to the Garnier-Robson [30] and Chou-Fasman [31] methods. The hydrophilicity, flexibility, surface accessibility, and antigenicity indices were also analyzed and predicted.

## Results

### Induced expression of the overlapping 16mer-peptides and western blot analysis of their antigenicity

The SDS-PAGE results showed that the 57 overlapping 16mer-peptides spanning the GTV Gc$^{1-549}$ fragment were successfully expressed under the 42˚C heat induction conditions (Fig 2A and 2C). The corresponding western blot analysis of their antigenicity showed that P3

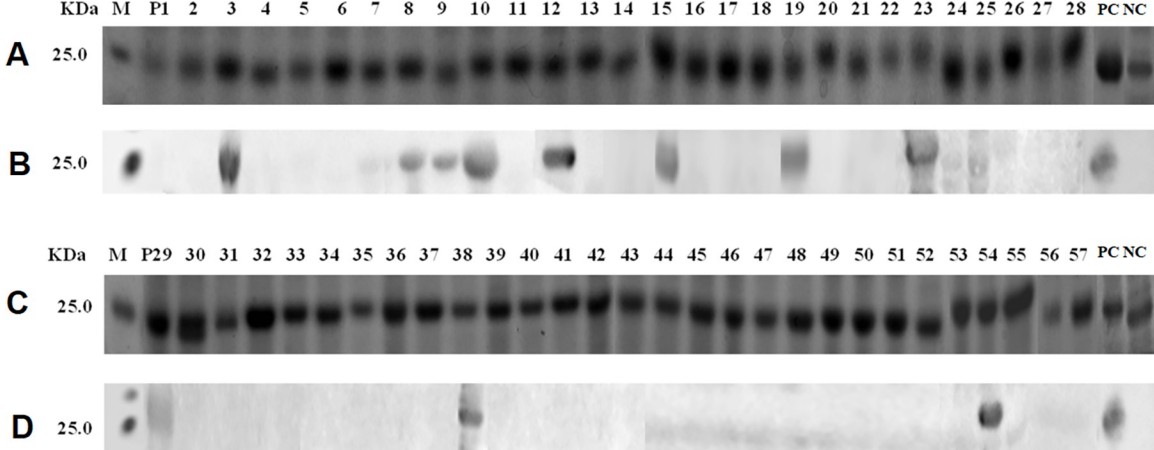

**Fig 2. SDS-PAGE and Western blot analysis of expressed 16mer-peptides.** (A and C) SDS-PAGE analysis of expressed 16mer-peptides. The numbers of P1-P57 indicate each 16mer-peptides in cell total proteins. The cell proteins of each recombinant clone were resolved by 12% SDS-PAGE and stained with Coomassie brilliant blue. M, the protein molecular marker; PC, Positive control of expressed 16mer peptide in GTV-NP; NC, Negative control of GST188 carrier protein expressed by pXXGST-2. (B and D) Western blot analysis for mapping reactive 16mer-peptides in P1-P57. The rabbit antiserum (1:300 dilution) against GTV-Gc was used in Western blot analysis. The reactive bands in Western blot analysis were visualized by enhanced chemiluminescence.

(Gc$^{17-32}$), P8 (Gc$^{57-72}$), P9 (Gc$^{65-80}$), P10 (Gc$^{73-89}$), P12 (Gc$^{89-104}$), P15 (Gc$^{113-128}$), P19 (Gc$^{145-160}$), P23 (Gc$^{177-192}$), P29 (Gc$^{225-240}$), P38 (Gc$^{297-312}$), and P54 (Gc$^{425-440}$) could bind specifically to the rabbit pAbs (Fig 2B and 2D) and were therefore potential linear antigenic epitope motifs.

## Induced expression and western blot analysis of the overlapping 8mer-peptides

To identify the minimal antigenic epitopes that could bind specifically to the rabbit pAbs, the eleven positive 16mer peptide fragments were further designed as 8mer-peptides with seven overlapping amino acids. The SDS-PAGE and sequencing results showed that all overlapping 8mer-peptides with a GST188 tag were correctly expressed in *E. coli*. Moreover, the western blot analysis confirmed that P9-3 (Gc$^{68-75}$) in the P9 epitope was specifically recognized by the rabbit pAbs, indicating that the minimal motif of the P9 epitope was "SSYYVPDA" (named EGc3). Similarly, in the P12, P15, P19, P23, and P54 epitopes, the minimal motifs recognized by the rabbit pAbs were found to be P12-1 (Gc$^{90-97}$), P12-7 (Gc$^{96-103}$), P15-1 (Gc$^{114-121}$), P15-2 (Gc$^{115-122}$), P19-2 (Gc$^{147-154}$), P19-3 (Gc$^{148-155}$), P23-1 (Gc$^{178-185}$), P23-2 (Gc$^{179-186}$), P54-3 (Gc$^{428-435}$), P54-4 (Gc$^{429-436}$), and P54-5 (Gc$^{430-437}$). Interestingly, none of the overlapping 8mer-peptides generated from the other five positive 16mer-peptides (i.e., P3, P8, P10, P29, and P38) showed a positive reaction, suggesting that their minimal epitope motifs might contain longer antigenic fragments. Finally, as shown in Fig 3, seven BCEs were identified, namely EGc3 ($^{68}$SSYYVPDA$^{75}$), EGc6 ($^{90}$DCQSGCPS$^{97}$), EGc7 ($^{96}$PSHFTSNS$^{103}$), EGc8 ($^{115}$AGLGFSG$^{121}$), EGc9 ($^{148}$ENPHGVI$^{154}$), EGc10 ($^{179}$KVFHPMS$^{185}$), and EGc13 ($^{430}$DIPRFV$^{435}$).

## Induced expression and western blot analysis of the overlapping 10mer-peptides

To identify the minimal antigenic epitope motifs of the positive 16mer-peptides P3, P8, P10, P29, and P38 that could bind specifically to rabbit pAbs, we designed them as 10mer-peptides with seven overlapping amino acid residues for validation. The SDS-PAGE and sequencing results showed that all overlapping 10mer-peptides with a GST188 tag were correctly expressed in *E. coli*, and western blot analysis showed that P3-9 (Gc$^{18-27}$), P3-10 (Gc$^{19-28}$), P8-8 (Gc$^{57-66}$), P8-9 (Gc$^{58-67}$), P10-14 (Gc$^{76-85}$), P10-17 (Gc$^{79-88}$), P29-11 (Gc$^{228-237}$), P29-12 (Gc$^{229-238}$), P29-13 (Gc$^{230-239}$), and P38-16 (Gc$^{303-312}$) were specifically recognized by the rabbit pAbs. The minimal motifs in these epitopes were EGc1 ($^{19}$KVCATTGRA$^{27}$), EGc2 ($^{58}$KKINLKCKK$^{66}$), EGc4 ($^{75}$ARSRCTSVRR$^{84}$), EGc5 ($^{79}$CTSVRRCRWA$^{88}$), EGc11 ($^{230}$QAGMGVVG$^{237}$), and EGc12 ($^{303}$RSHDSQGKIS$^{312}$) (Fig 4). Thus, a total of 13 minimal antigenic epitope motifs were identified in GTV Gc$^{1-549}$. The overlapping 16/10/8mer peptide sequences and their corresponding locations on the Gc protein are shown in Fig 5.

## Validation of the antigenicity of the peptides using GTV-positive sheep sera

To investigate whether BCE screening with a rabbit anti-GTV Gc polyclonal antiserum could detect the IgG antibodies present naturally in livestock, the 13 short peptides of BCEs identified in this experiment were analyzed by western blot analysis using the positive serum of a sheep naturally infected with GTV and the negative serum of an uninfected sheep. The results showed that all 13 short peptides containing BCEs were correctly expressed (Fig 6A). Among the 13 minimal BCE motifs identified, only EGc2, EGc3, EGc4, EGc9, EGc10, EGc11, and

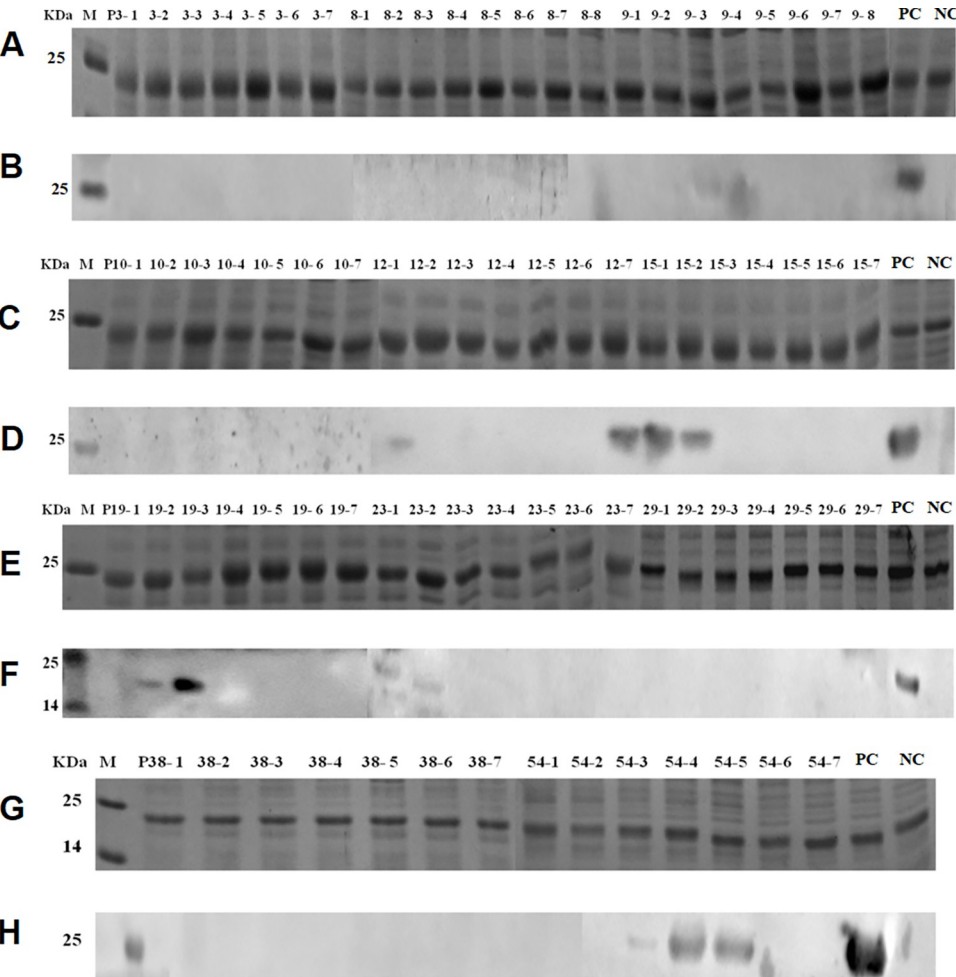

**Fig 3. SDS-PAGE and Western blot analysis of expressed 8mer-peptides.** (A, C, E and G) SDS-PAGE analysis of expressed 8mer-peptides. It indicates each 8mer peptide of P3, P8, P9, P10, P12, P15, P19, P23, P29, P38 and P54. The cell proteins of each recombinant clone were resolved by 12% SDS-PAGE and stained with Coomassie brilliant blue. M, the protein molecular marker. NC, Negative control of GST188 protein. PC, Positive control of mapped reactive P3. (B, D, F and H) Western blot analysis for mapping fine epitopes in each reactive 8-mer-peptides. The rabbit antiserum against GTV-Gc (1:300 dilution) was used in Western blot analysis. The reactive bands in Western blot analysis were visualized by enhanced chemiluminescence.

EGc12 were specifically recognized by the positive sheep serum (Fig 6B). None of the BCEs identified could be recognized by the negative sheep serum (Fig 6C). The Western blot results showed that there was a good consistency between the immune response of rabbit and sheep to GTV GC protein.

## Sequence conservation analysis and three-dimensional modeling

Using the Gendoc software, the amino acid sequences of the 13 BCEs were aligned with those of homologous proteins from 10 genetically closely related SFTSV strains from different regions (Japan, Accession no. AB817987.1; Japan, Accession no. AB985297.1; China, Accession no. HQ141605.1; China, Accession no. HQ830167.1; China, Accession no. HQ830167.1; China, Accession no. JQ670931.1; China, Accession no. KC505127.1; South Korea, Accession no. KF358692.1; China, Accession no. KP202164.1; China, Accession no. KX302598.1; and

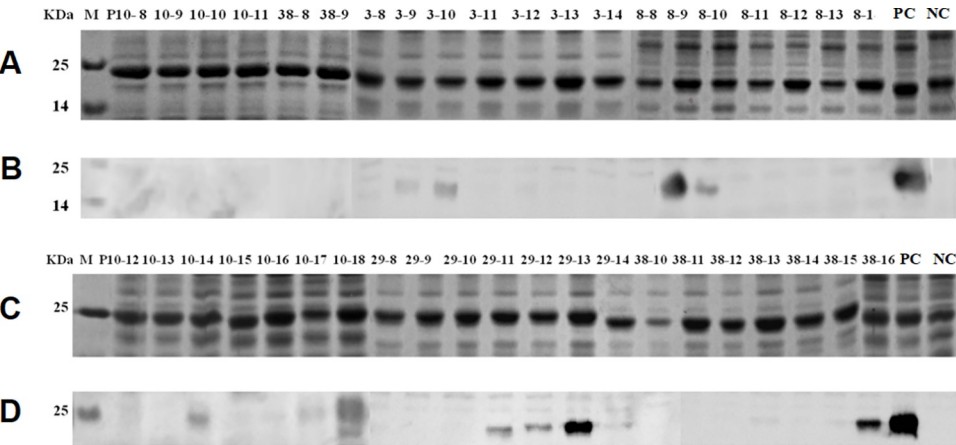

**Fig 4. SDS-PAGE and Western blot analysis of expressed 10mer-peptides.** (A and C) SDS-PAGE analysis of expressed 10mer-peptides for P3, P8, P10, P29 and P38. The cell proteins of each recombinant clone were resolved by 12% SDS-PAGE and stained with Coomassie brilliant blue. M, the protein molecular marker; NC, Negative control of GST188 carrier protein. PC, Positive control of P3. (B and D) Western blot analysis for mapping fine epitopes in each reactive 10mer peptide. The rabbit antiserum against GTV-Gc (1:300 dilution) was used in Western blot analysis. The reactive bands in Western blot analysis were visualized by enhanced chemiluminescence.

South Korea, Accession no. KY789438.1). The results revealed that EGc4 ($^{75}$ARSRCTSVRR$^{84}$), EGc5 ($^{79}$CTSVRRCRWA$^{88}$), EGc8 ($^{115}$AGLGFSG$^{121}$), EGc11 ($^{230}$QAGMGVVG$^{237}$), and EGc12 ($^{303}$RSHDSQGKIS$^{312}$) were identical among the representative SFTSV strains and showed 100% conserved sequences (Fig 7). Thus, they can be used as candidates for broad-spectrum multi-epitope vaccine design.

As shown in Fig 7, there was variability between the other epitopes and the SFTSV sequences. The sequence similarities of EGc1, EGc2, EGc3, EGc6, EGc7, EGc9, EGc10, and EGc13 were 87.50%, 87.50%, 87.50%, 87.50%, 87.50%, 85.71%, 71.43%, and 71.43%, respectively. In EGc1 ($^{19}$KVCATTGRA$^{27}$), "V" was changed to "E" at position Gc20; in EGc2 ($^{58}$KKINLKCKK$^{66}$), "V" was changed to "R" at position Gc59; in EGc3 ($^{68}$SSYYVPDA$^{75}$), "Y" was changed to "F" at position Gc71; in EGc6 ($^{90}$DCQSGCPS$^{97}$) and EGc7 ($^{96}$PSHFTSNS$^{103}$), "S" was changed to "P" at position Gc97; in EGc9 ($^{148}$ENPHGVI$^{154}$), "V" was changed to "I" at position Gc153; in EGc10 ($^{179}$KVFHPMS$^{185}$), "K" was changed to "V" and "R" was changed to "T" at positions Gc179 and Gc180; and in EGc13 ($^{430}$DIPRFV$^{435}$), "I" was changed to "R" and "V" was changed to "K" at positions Gc431 and Gc433.

The 3D structure of the Gc protein of GTV strain DXM was modeled using the Python 2.7 software, where different color markers were used to locate the BCEs in the model. As shown in Fig 8B, all the identified antigenic epitopes were located on the surface of the Gc protein. Moreover, the 13 BCEs were located at positions that were easily accessible for antibody chimerization and thus could facilitate the binding of specific antibodies.

The secondary structure of Gc$^{1-549}$ was predicted using the DNASTAR Protean software (DNASTAR Inc., Madison, WI, USA), which showed that the identified epitopes formed part of the β-sheet and α-helix regions (Fig 8B). Moreover, the identified epitopes were predicted to be located on the surface of the GTV Gc protein with a high antigenic index and hydrophilicity, suggesting that they might be important BCEs of this glycoprotein (Fig 8A). The 3D modeling and protein secondary structure analysis indicated that the predictions were consistent with the identified epitopes and that the identified BCEs were all located in the hydrophilic region.

| Peptide items | Amino acids | Position in Gc | Peptide items | Amino acids | Position in Gc |
|---|---|---|---|---|---|
| P1 | L V H A D S K L I S C K Q G G N | 1-16 | P19-6 | H G V I W K V S | 151-158 |
| P2 | I S C K Q G G N N N K V C A T T | 9-24 | P19-7 | G V I W K V S P | 152-159 |
| P3 | N N K V C A T T G R A L L P A V | 17-32 | P20 | V I W K V S P C A A W V P S A E | 153-168 |
| P3-8 | N N K V C A T T G R | 17-26 | P21 | A A W V P S A E V E V T L P S G | 161-176 |
| P3-9 | N K V C A T T G R A (EGc1) | 18-27 | P22 | V E V T L P S G K S K V F H P M | 169-184 |
| P3-10 | K V C A T T G R A L | 19-28 | P23 | K S K V F H P M S G V P T Q A F | 177-192 |
| P3-11 | V C A T T G R A L L | 20-29 | P23-1 | S K V F H P M S (EGc10) | 178-185 |
| P3-12 | C A T T G R A L L P | 21-30 | P23-2 | K V F H P M S G | 179-186 |
| P3-13 | A T T G R A L L P A | 22-31 | P23-3 | V F H P M S G V | 180-187 |
| P3-14 | T T G R A L L P A V | 23-32 | P23-4 | F H P M S G V P | 181-188 |
| P4 | G R A L L P A V N P G Q T A C L | 25-40 | P23-5 | H P M S G V P T | 182-189 |
| P5 | N P G Q T A C L H F S A P G S P | 33-48 | P23-6 | P M S G V P T Q | 183-190 |
| P6 | H F S A P G S P D S K C L K I K | 41-56 | P23-7 | M S G V P T Q A | 184-191 |
| P7 | D S K C L K I K V K K I N L K C | 49-64 | P24 | S G V P T Q A F K G V S I T Y L | 185-200 |
| P8 | V K K I N L K C K K A S S Y Y V | 57-72 | P25 | K G V S I T Y L G S E L E V S G | 193-208 |
| P8-9 | V K K I N L K C K K | 57-66 | P26 | G S E L E V S G L T E L C E I E | 201-216 |
| P8-10 | K K I N L K C K K A (EGc2) | 58-67 | P27 | L T E L C E I E E L K S G R L A | 209-224 |
| P8-10 | K I N L K C K K A S | 59-68 | P28 | E L K S G R L A L A P C N Q A G | 217-232 |
| P8-10 | I N L K C K K A S S | 60-69 | P29 | L A P C N Q A G M G V V G K I G | 225-240 |
| P8-10 | N L K C K K A S S Y | 61-70 | P29-8 | L A P C N Q A G M G | 225-234 |
| P8-10 | L K C K K A S S Y Y | 62-71 | P29-9 | A P C N Q A G M G V | 226-235 |
| P8-10 | K C K K A S S Y Y V | 63-72 | P29-10 | P C N Q A G M G V V | 227-236 |
| P9 | K K A S S Y Y V P D A R S R C T | 65-80 | P29-11 | C N Q A G M G V V G | 228-237 |
| P9-1 | K A S S Y Y V P | 66-73 | P29-12 | N Q A G M G V V G K (EGc11) | 229-238 |
| P9-2 | A S S Y Y V P D | 67-74 | P29-13 | Q A G M G V V G K I | 230-239 |
| P9-3 | S S Y Y V P D A (EGc3) | 68-75 | P29-14 | A G M G V V G K I G | 231-240 |
| P9-4 | S Y Y V P D A R | 69-76 | P30 | M G V V G K I G E I Q C S S E E | 233-248 |
| P9-5 | Y Y V P D A R S | 70-77 | P31 | E I Q C S S E E S A R T I K K D | 241-256 |
| P9-6 | Y V P D A R S R | 71-78 | P32 | S A R T I K K D G C I W N S D L | 249-264 |
| P9-7 | V P D A R S R C | 72-79 | P33 | G C I W N S D L V G I E L R V D | 257-272 |
| P9-8 | P D A R S R C T | 73-80 | P34 | V G I E L R V D D A V C F S K I | 265-280 |
| P10 | P D A R S R C T S V R R C R W A | 73-88 | P35 | D A V C F S K I T S V E A V A N | 273-288 |
| P10-8 | P D A R S R C T S V | 73-82 | P36 | T S V E A V A N Y S A I P T I I | 281-296 |
| P10-9 | D A R S R C T S V R | 74-83 | P37 | Y S A I P T I I G G L R F E R S | 289-304 |
| P10-10 | A R S R C T S V R R (EGc4) | 75-84 | P38 | G G L R F E R S H D S Q G K I S | 297-312 |
| P10-11 | R S R C T S V R R C | 76-85 | P38-8 | G G L R F E R S H D | 297-306 |
| P10-12 | S R C T S V R R C R | 77-86 | P38-9 | G L R F E R S H D S | 298-307 |
| P10-13 | R C T S V R R C R W | 78-87 | P38-10 | L R F E R S H D S Q | 299-308 |
| P10-14 | EGc5 — C T S V R R C R W A | 79-88 | P38-11 | R F E R S H D S Q G | 300-309 |
| P11 | S V R R C R W A G D C Q S G C P | 81-96 | P38-12 | F E R S H D S Q G K | 301-310 |
| P12 | G D C Q S G C P S H F T S N S F | 89-104 | P38-13 | E R S H D S Q G K I | 302-311 |
| P12-1 | D C Q S G C P S — EGc6 | 90-97 | P38-14 | EGc12 — R S H D S Q G K I S | 303-312 |
| P12-2 | C Q S G C P S H | 91-98 | P39 | H D S Q G K I S G S P L D I T A | 305-320 |
| P12-3 | Q S G C P S H F | 92-99 | P40 | G S P L D I T A I R G E F S V S | 313-328 |
| P12-4 | S G C P S H F T | 93-100 | P41 | I R G E F S V S Y R G L R L S L | 321-336 |
| P12-5 | G C P S H F T S | 94-101 | P42 | Y R G L R L S L S E I T A T C T | 329-344 |
| P12-6 | C P S H F T S N | 95-102 | P43 | S E I T A T C T G E V T N I S G | 337-352 |
| P12-7 | EGc7 — P S H F T S N S | 96-103 | P44 | G E V T N I S G C Y S C M M G A | 345-360 |
| P13 | S H F T S N S F S D D W A G K M | 97-112 | P45 | C Y S C M M G A K V S I R L H S | 353-368 |
| P14 | S D D W A G K M D R A G L G F S | 105-120 | P46 | K V S I R L H S N K N S T A H L | 361-376 |
| P15 | D R A G L G F S G C S D G C G G | 113-128 | P47 | N K N S T A H L K C S S D E T A | 369-384 |
| P15-1 | R A G L G F S G (EGc8) | 114-121 | P48 | K C S S D E T A F S V S E G V H | 377-392 |
| P15-2 | A G L G F S G C | 115-122 | P49 | F S V S E G V H S Y T V S L S Y | 385-400 |
| P15-3 | G L G F S G C S | 116-123 | P50 | S Y T V S L S Y D H A V V D E T | 393-408 |
| P15-4 | L G F S G C S D | 117-124 | P51 | D H A V V D E T C I L N C G G H | 401-416 |
| P15-5 | G F S G C S D G | 118-125 | P52 | C I L N C G G H E S Q V N V K G | 409-424 |
| P15-6 | F S G C S D G C | 119-126 | P53 | E S Q V N V K G N L V F L D I P | 417-432 |
| P15-7 | S G C S D G C G | 120-127 | P54 | N L V F L D I P R F V D G S Y V | 425-440 |
| P16 | G C S D G C G G A A C G C F N A | 121-136 | P54-1 | L V F L D I P R | 426-433 |
| P17 | A A C G C F N A A P S C I F W R | 129-144 | P54-2 | V F L D I P R F | 427-434 |
| P18 | A P S C I F W R K W V E N P H G | 137-152 | P54-3 | F L D I P R F V | 428-435 |
| P19 | K W V E N P H G V I W K V S P C | 145-160 | P54-4 | L D I P R F V D (EGc13) | 429-436 |
| P19-1 | W V E N P H G V | 146-153 | P54-5 | D I P R F V D G | 430-437 |
| P19-2 | V E N P H G V I (EGc9) | 147-154 | P54-6 | I P R F V D G S | 431-438 |
| P19-3 | E N P H G V I W | 148-155 | P54-7 | P R F V D G S Y | 432-439 |
| P19-4 | N P H G V I W K | 149-156 | P55 | R F V D G S Y V Q T Y H S T V P | 433-448 |
| P19-5 | P H G V I W K V | 150-157 | P56 | Q T Y H S T V P T G A S I P S P | 441-456 |
|  |  |  | P57 | T G A S I P S P T D | 449-459 |

**Fig 5. The bio-synthetic overlapping peptides derived from IgG-reactive peptides of Gc$^{1-459}$.** The yellow highlighting represents the common sequences among immunodominant peptides that react with pAbs according to western blot analysis.

## Discussion

*Bunyavirales* are the largest order of RNA viruses with the ability to infect a wide range of hosts, including humans, arthropods, and plants [32]. Several emerging bunyaviruses, such as the Rift Valley fever virus [33], Crimean-Congo hemorrhagic fever virus [34], HRTV [7], and SFTSV [35], have been shown to pose a serious threat to human health. SFTSV can be transmitted from ticks to humans, with human-to-human transmission occurring in rare instances [36–38], and can lead to the manifestation of diseases characterized by fever, gastrointestinal symptoms, leukopenia, and thrombocytopenia. SFTSV and its related viruses are important emerging pathogens for which there are currently no specific antiviral drugs or vaccines [32]. As a novel bunyavirus, GTV belongs to the genus *Banyangvirus* together with HRTV and SFTSV [5]. Shen et al. [6]. performed in vitro and in vivo experiments to characterize the infection properties of GTV in human- and animal-derived cells. They found that infection of C57BL/6 mice with GTV could cause viremia or organopathological changes, indicating that this virus was a potential pathogen capable of infecting humans and animals. Serological testing of local residents in the Guertu area of Wusu City, Xinjiang revealed that 19.8% were positive for anti-GTV antibodies, and three of them had neutralizing antibody activity against the virus, which further suggested the potential risk of this pathogen to human health.

Studies have shown that viral glycoproteins are the targets of neutralizing antibodies and that the envelope glycoproteins are responsible for the binding of the virus particles to cellular receptors and subsequent viral fusion [39]. In viral infections, almost all neutralizing polyclonal or monoclonal antibodies are specific to the Gn and Gc glycoproteins [40, 41]. Therefore, the identification of BCE domains in GTV glycoproteins will provide new targets for the design and development of vaccines and diagnostic reagents for this virus. Although several studies on the identification of BCEs of the GTV Gn protein have been reported [17], there are no published studies on epitope mapping of the GTV Gc protein or of its IgG motif recognition sites.

In this study, an improved BSP method was used to map linear BCEs spanning the full length of the GTV Gc protein. Using the overlapping BSP method, the Gc protein was divided

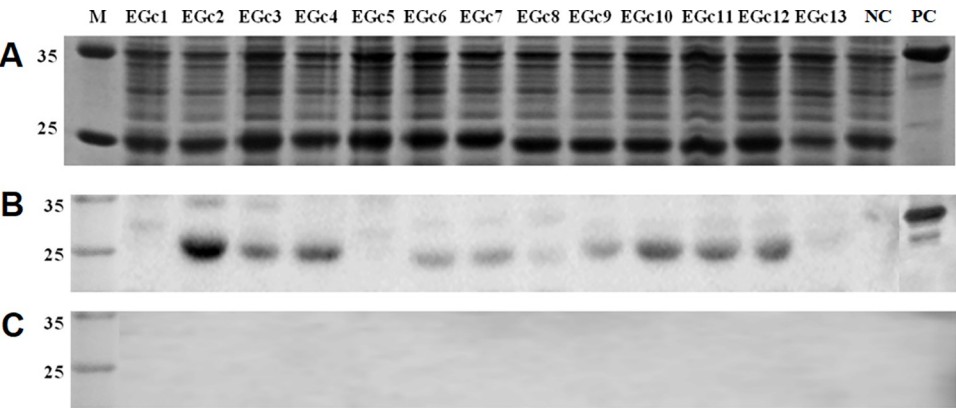

**Fig 6. Western blot analysis of mapped BCEs using sheep sera.** (A) SDS-PAGE analysis of expressed BCEs. (B) Using a positive serum from a sheep confirmed GTV-infection. (C) Using a serum from healthy sheep with no history of GTV infection as a negative control. PC, Positive control of GTV-Gc1.

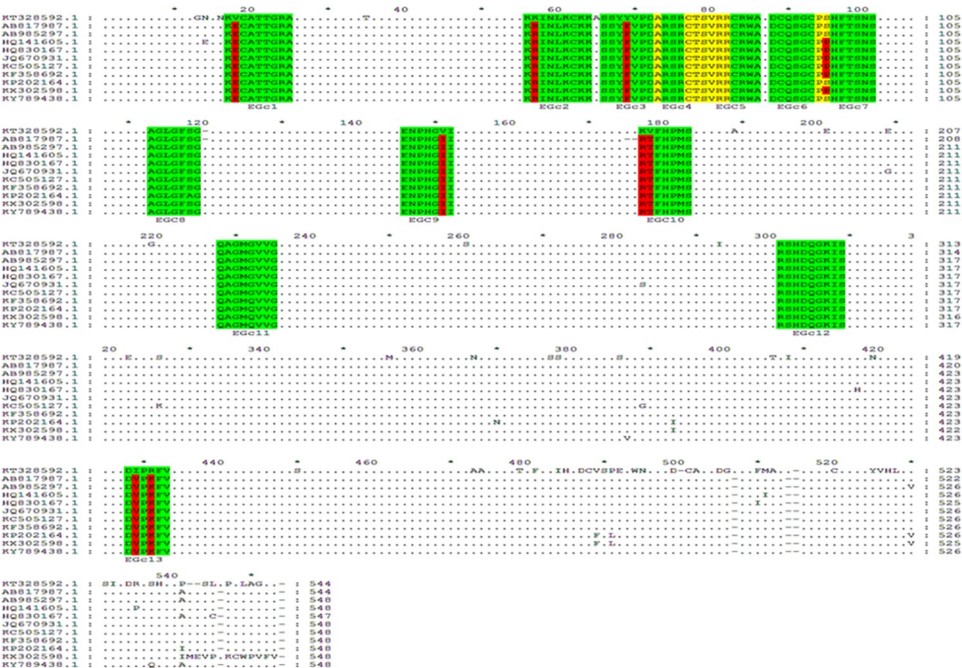

**Fig 7. Sequence comparison between GTV-Gc$^{1-549}$ and 10 SFTSV strains.** The GenBank codes and sources are shown at left and the sequence analysis was based on the ClustalW program. The nine of fine BCEs and APs recognized by pAbs are highlighted, and the variable aa residue within the BCE motif are highlighted in red. Dots (.) indicate identical aa residue in ten SFTSV strains (For interpretation of the references to color in this figure legend, the reader is referred to the web version of this article).

into 57 overlapping 16mer-peptides. Of these, 11 positive 16mer-peptides were selected through western blot analysis using rabbit anti-GTV-Gc pAbs and were further divided into 79 overlapping 8mer-peptides and 35 overlapping 10mer-peptides. Using immunoblot analysis, the following 13 antigenic epitope motifs were finally identified: EGc1 ($^{19}$KVCATTGRA$^{27}$), EGc2 ($^{58}$KKINLKCKK$^{66}$), EGc3 ($^{68}$SSYYVPDA$^{75}$), EGc4 ($^{75}$ARSRCTSVRR$^{84}$), EGc5 ($^{79}$CTSVRRCRWA$^{88}$), EGc6 ($^{90}$DCQSGCPS$^{97}$), EGc7 ($^{96}$PSHFTSNS$^{103}$), EGc8 ($^{115}$AGLGFSG$^{121}$), EGc9 ($^{148}$ENPHGVI$^{154}$), EGc10 ($^{179}$KVFHPMS$^{185}$), EGc11 ($^{230}$QAGMGVVG$^{237}$), EGc12 ($^{303}$RSHDSQGKIS$^{312}$), and EGc13 ($^{430}$DIPRFV$^{435}$). Amino acid sequence alignment of the BCEs with homologous proteins from 10 closely related SFTSV strains showed that EGc4, EGc5, EGc8, EGc11, and EGc12 were the most conserved epitopes (i.e., with 100% sequence identity) and could be used as candidates for the future design of broad-spectrum multi-epitope vaccines against GTV and SFTSV. EGc1, EGc2, EGc3, EGc6, EGc7, EGc9, and EGc13 differed more significantly from the SFTSV Gc sequences, showing differences of 1–2 amino acids, and thus could be used for the serological verification of GTV infection in patients in the future. Since no GTV-infected patients have been identified thus far, it is not possible to confirm whether these epitopes are specific to this virus. In the future, detection of serum antibodies to SFTSV and GTV simultaneously in patients will remain necessary to compare the specificity and sensitivity of these epitopes and to establish a theoretical basis for the development of broad-spectrum and specific detection methods and vaccines for bunyaviruses.

Additionally, the 13 BCEs identified in this study were all confirmed by secondary and 3D structural analyses to be located on the surface of the protein structure and showed good antigenicity. The prediction results included most of the experimentally identified epitope motifs,

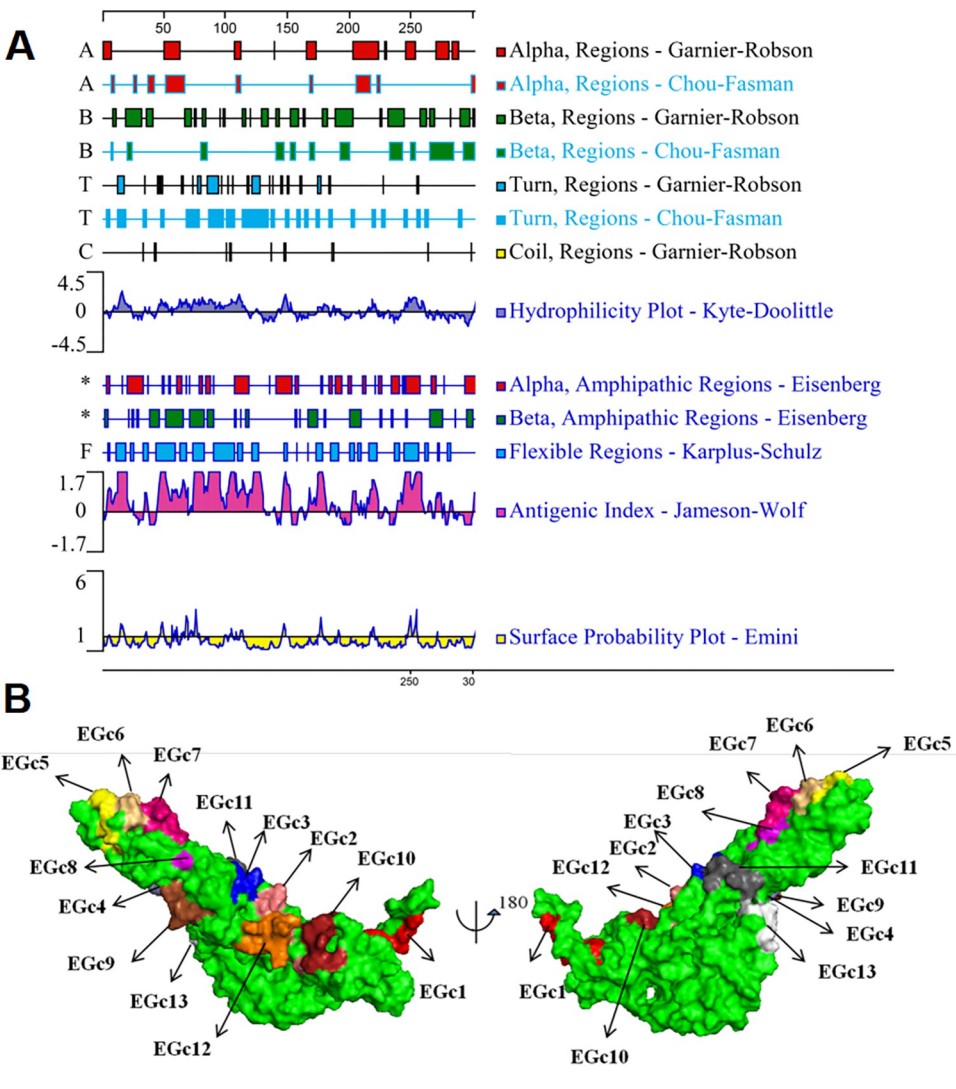

**Fig 8. Prediction of Gc secondary structure and 3D localization of each BCEs.** (A) Epitope prediction for GTV-Gc using DNAStar-Protean software. The secondary structure, flexibility plot, hydrophilicity, surface probability, and antigenicity index for GTV-Gc were taken into consideration. (B) Location distribution on 3D structure of mapped BCEs and APs on molecular surface were shown in different colors. EGc1 (red), EGc2 (salmon), EGc3 (blue), EGc4 (slate), EGc5 (yellow), EGc6 (wheat), EGc7 (hot pink), EGc8 (light magenta), EGc9 (brown),EGc10 (firebrick), EGc11 (gray), EGc12 (orange), EGc13 (white). The figures were generated using the PyMOL™ molecular graphics system (For interpretation of the references to colour in this figure legend, the reader is referred to the web version of this article).

indicating that the epitope prediction tool combined with the BSP method was a reliable approach for epitope identification and mapping and could reduce the experimental workload and costs of epitope mapping and immunodiagnosis [14, 26, 27]. Moreover, none of the 13 BCEs required purification, mainly because the recombinant fusion peptide expressed by the pXXGST-3 vector was larger than the protein in the empty vector, and the expression products were located in the weak antigenic region of *E. coli*. Therefore, the expression products could be used directly for western blot analysis without a purification step, thereby reducing the time needed for screening of the recombinant bacteria and the workload [23, 24].

Among study limitations, this study only identified linear epitopes of the Gc protein. Also, there are currently no B cell epitopes predicted to be affected by different MHC haplotypes.

The mechanism of virus neutralization by anti-Gc protein antibodies remains unknown, and neutralization through identified linear epitopes of Gc protein has not been characterized. Notably, improved BSP method adopted is only suitable for screening simple linear epitopes. In the future, peptide microarray high-throughput screening, mass spectrometry, and other techniques may be necessary to identify conformational epitopes in order to avoid missing the detection of specific epitopes of SFTSV and GTV that could be used to distinguish the two viruses. In the future, serological validation of GTV should be conducted to investigate whether these epitopes are specific to this virus and whether they can be used to detect SFTSV and GTV antibodies or to design broad-spectrum vaccines.The epitopes of the GTV Gc protein obtained in this study can serve as fundamental data for future research works aimed at elucidating the immunological properties of the GTV glycoproteins and for developing multi-epitope diagnostic reagents and vaccines.

## Supporting information

**S1 Raw images.**
(PDF)

## Author Contributions

**Conceptualization:** Meilipaiti Yusufu, Ayipairi Abula, Juntao Ding.

**Data curation:** Meilipaiti Yusufu, Ayipairi Abula, Boyong Jiang.

**Formal analysis:** Ayipairi Abula, Boyong Jiang, Jiayinaguli Zhumabai, Yijie Li.

**Investigation:** Jiayinaguli Zhumabai, Fei Deng, Yijie Li, Yujiang Zhang, Juntao Ding, Surong Sun.

**Methodology:** Meilipaiti Yusufu, Ayipairi Abula, Boyong Jiang.

**Project administration:** Fei Deng.

**Software:** Meilipaiti Yusufu, Ayipairi Abula, Boyong Jiang, Yijie Li, Yujiang Zhang, Surong Sun.

**Supervision:** Jiayinaguli Zhumabai, Fei Deng, Yijie Li, Yujiang Zhang, Juntao Ding, Surong Sun.

**Validation:** Meilipaiti Yusufu.

**Visualization:** Boyong Jiang, Jiayinaguli Zhumabai, Juntao Ding.

**Writing – original draft:** Meilipaiti Yusufu, Ayipairi Abula, Juntao Ding.

**Writing – review & editing:** Fei Deng, Yujiang Zhang, Surong Sun.

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
