## [Decision Letter · Decision Letter 0]

18 May 2021

PONE-D-21-02919

Fine mapping of the antigenic epitopes of the Gc protein of Guertu virus

PLOS ONE

Dear Dr. Sun,

Thank you for submitting your manuscript to PLOS ONE. After careful consideration, we feel that it has merit but does not fully meet PLOS ONE’s publication criteria as it currently stands. Therefore, we invite you to submit a revised version of the manuscript that addresses the points raised during the review process.

We look forward to receiving your revised manuscript.

Kind regards,

Florian Krammer, PhD

Academic Editor

PLOS ONE

Journal Requirements:

[This work was supported by Science Foundation of China (No.81690369,81760365 to S.R.S.), and the Science Research Key Project of Xinjiang Education Department (No.XJEDU2019I002 to S.R.S.).The funders had no role in study design, data collection and analysis, decision topublish, or preparation of the manuscript.]

 [This work was supported partly by grants from the National Natural Science.]

4. We noticed you have some minor occurrence of overlapping text with the following previous publication(s), which needs to be addressed:

- https://journals.plos.org/plosone/article?id=10.1371/journal.pone.0223978

- https://pubmed.ncbi.nlm.nih.gov/31618247/

The text that needs to be addressed involves the Results section.

In your revision ensure you cite all your sources (including your own works), and quote or rephrase any duplicated text outside the methods section. Further consideration is dependent on these concerns being addressed.

Reviewers' comments:

Reviewer's Responses to Questions

**Comments to the Author**

1. Is the manuscript technically sound, and do the data support the conclusions?

Reviewer #1: Yes

Reviewer #2: Yes

2. Has the statistical analysis been performed appropriately and rigorously? 

Reviewer #1: Yes

Reviewer #2: N/A

3. Have the authors made all data underlying the findings in their manuscript fully available?

Reviewer #1: Yes

Reviewer #2: Yes

4. Is the manuscript presented in an intelligible fashion and written in standard English?

Reviewer #1: Yes

Reviewer #2: Yes

5. Review Comments to the Author

Reviewer #1: Yusufu et al describe linear BCEs of the Guertu virus Gc proteins using polyclonal serum they generated in animals.

While the work done is important, there is not much relevance to the real world and infection in humans. The authors did find BCEs which is not surprising because the serum was polyclonal. In addition, numerous studies are needed to see if neutralization is crucial at these epitopes or not. 

The authors should discuss in detail in the discussion section how their work will lead to new discoveries in the field and how they will move forward. How does this work relate to vaccine design? What are the lessons learned?

Line 74: The authors should discuss if this virus is a BSL2 or BSL3 pathogen and the limitations that exist in studying this virus in the labLine 89: The authors should discuss receptors for entry and other entry factors known.

Line 90: It is unclear for a naive reader whether its Gc or Gn that mediates fusion or entry?Line 100-112: Discussion of methodology used seems excessive. It may be easier to just discuss the benefits of the approach used in the paper.

Line 130: Since this method only allows for elucidation of linear BCEs, are the authors concerned that the majority of the novel neutralizing epitopes may be overlooked? A proper folded and stable glycoprotein is key to eliciting neutralizing antibodies in terms of vaccine design.

Line 432: Does disease caused by the virus vary from person to person?

Reviewer #2: Yusufu M et. al provided a study on fine epitope mapping of the Gc glycoprotein of Guertu virus.

They were able to identify 13 B-cell epitopes, of whom 7 could be recognized by GTV IgG positive sheep sera.

Moreover, they could show that BCE´s motifs are highly conserved among 10 SFTSV strains from different countries and lineages.

Altogether, this study is an interesting work on elucidation of immunologic properties of GTV glycoproteins and expands the knowledge in this field.

6. PLOS authors have the option to publish the peer review history of their article (what does this mean?). If published, this will include your full peer review and any attached files.

Reviewer #1: No

Reviewer #2: No

---

## [Author Response · Author response to Decision Letter 0]

1 Jul 2021

Reviewer #1: Yusufu et al describe linear BCEs of the Guertu virus Gc proteins using polyclonal serum they generated in animals.

While the work done is important, there is not much relevance to the real world and infection in humans. The authors did find BCEs which is not surprising because the serum was polyclonal. In addition, numerous studies are needed to see if neutralization is crucial at these epitopes or not.

The authors should discuss in detail in the discussion section how their work will lead to new discoveries in the field and how they will move forward. How does this work relate to vaccine design? What are the lessons learned?

Response: Thanks for the good remind. Since GTV is a virus that requires BSL2 laboratory operation, the use of attenuated vaccine in vaccine design may cause biosafety risks, and GTV glycoprotein Gn and Gc are large and can not be expressed in tandem, it is difficult to operate in vaccine design. The epitope of GTV Gn has also been identified, the B cell linear epitopes on Gn and Gc may be the epitope recognized by GTV neutralizing antibody. In that case, we will make a safe and convenient multi-epitope vaccine that easy to operate all the B cell linear epitopes after tandem expression. It can be used as a new target for GTV vaccine design.

Line 74: The authors should discuss if this virus is a BSL2 or BSL3 pathogen and the limitations that exist in studying this virus in the lab Line 89: The authors should discuss receptors for entry and other entry factors known.

Response: Thanks for the good remind. This virus is a BSL2 pathogen and the limitations that exist in studying this virus in the lab. According to your suggestion, we added “At present, the virus cell receptor binding domain of SFTSV, which is most closely related to GTV evolution, is located on glycoprotein Gc [11]. DC-SIGN has been identified as the receptor factor on the host cell membrane [12]. Some studies have also shown that HS and NMMHC-IIA may also be the entry factors related to cell entry of SFTSV [13].” in the introduction on section in Line 65-68.

Line 90: It is unclear for a naive reader whether its Gc or Gn that mediates fusion or entry?Line 100-112: Discussion of methodology used seems excessive. It may be easier to just discuss the benefits of the approach used in the paper.

Response: Thanks for the kind remind and suggestion.At present, the virus cell receptor binding domain of SFTSV, which is most closely related to GTV evolution, is located on glycoprotein Gc. According to your suggestion, we deleted other methods for identifying B cell epitopes that are not related to this study from the original manuscript, and we just discuss the benefits of the approach used in the paper.

Line 130: Since this method only allows for elucidation of linear BCEs, are the authors concerned that the majority of the novel neutralizing epitopes may be overlooked? A proper folded and stable glycoprotein is key to eliciting neutralizing antibodies in terms of vaccine design.

Response: Thanks for the good remind. Improved BSP method adopted is only suitable for screening simple linear epitopes. In the future, peptide microarray high-throughput screening, mass spectrometry, and other techniques may be necessary to identify conformational epitopes in order to avoid missing the detection of specific epitopes of SFTSV and GTV that could be used to distinguish the two viruses.

Line 432: Does disease caused by the virus vary from person to person?

Response: Thanks for the kind remind. There is no case of GTV infection, but the genetic relationship between GTV and SFTSV is similar in Philadelphia. It is predicted that the disease caused by GTV may have symptoms similar to SFTSV.

Reviewer #2: Yusufu M et. al provided a study on fine epitope mapping of the Gc glycoprotein of Guertu virus.They were able to identify 13 B-cell epitopes, of whom 7 could be recognized by GTV IgG positive sheep sera.Moreover, they could show that BCE´s motifs are highly conserved among 10 SFTSV strains from different countries and lineages.Altogether, this study is an interesting work on elucidation of immunologic properties of GTV glycoproteins and expands the knowledge in this field.

---

## [Editor Report · Decision Letter 1]

29 Nov 2021

PONE-D-21-02919R1Fine mapping of the antigenic epitopes of the Gc protein of Guertu virusPLOS ONE

Dear Dr. Sun,

Thank you for submitting your revised manuscript to PLOS ONE. We invite you to submit a revised version of the manuscript that addresses the points raised during the review process.

We have tried to contact the reviewers who reviewed the original version of your manuscript in the past weeks. However, after many attempts, we failed to get their responses. We were unable to obtain additional reviewers to review your revised manuscript. This happens these days during the pandemic, a difficult time for many, and we apologize for the delayed process of your revised manuscript.

There are several issues to be addressed:

In your point-by-point response to reviewer’s comments, you need to indicate whether you’ve revised your manuscript accordingly, and how and where.Expression for lack of epitope response in sheep serum: Amino acid sequences of B-cell epitopes are not universal among species. Epitopes need to be processed by proteases and recognized by the MHC complex for presentation to the surface of cell membrane in antigen presenting cells. Differences in protease processing and MHC presentation of antigen epitopes (either B or T cell epitope) exist naturally among different species of animals. Your explanation does not make much sense (Lines 279-81).Figure 7 & 8 are missing. Somehow we can’t find them.Carefully edit the manuscript to remove errors in spelling, expression, and gramma. 

Minor Issues:

There are many errors with spelling, expression, or English gramma. These errors need to be corrected. Please carefully edit the manuscript before resubmission. For example: 

Line 66: “… which is most closely related to GTV 66 evolution”

Line 77: “… are as no…”

Line 144: “…the GTV Gc protein was designed as fifty-seven 16-mer peptides…”

Line 208: 16mer-peptide or 16mer peptide, with or without hyphen? Be consistent in the text and

all figure legends. Please check.

Line 209: SDS-PAGE gel electrophoresis: GE in PAGE represents gel electrophoresis, No need to repeat.

Correct this in all figure legends. Please check.

Line 210: blue.M :You need a space between the two words (letters). There are several other places

with similar spelling errors in the text or figure legends. Please check.

Line 212: Western blot (212), Western blotting (213, 214), western blot analysis (221), or western

blotting analysis (269)? Be consistent in the text and all figure legends. Please check.

Line 209: What is r-clone?

Line 253: EGc11 and EGc12 should not have the same aa numbers from 230 to 237.

Line 259: expressed10mer-peptides

Line 259: “It indicates each short peptide for numbers 10mer…” Correct expression?

Page numbers must be inserted on each page for submitted manuscripts. ==============================

We look forward to receiving your revised manuscript.

Kind regards,

Zheng Xing

Academic Editor

PLOS ONE
---

## [Author Response · Author response to Decision Letter 1]

29 Dec 2021

1.In your point-by-point response to reviewer’s comments, you need to indicate whether you’ve revised your manuscript accordingly, and how and where.

Expression for lack of epitope response in sheep serum: Amino acid sequences of B-cell epitopes are not universal among species. Epitopes need to be processed by proteases and recognized by the MHC complex for presentation to the surface of cell membrane in antigen presenting cells. Differences in protease processing and MHC presentation of antigen epitopes (either B or T cell epitope) exist naturally among different species of animals. Your explanation does not make much sense (Lines 279-81).

Response: Sorry, a paragraph in lines 279-281 is not accurate enough and has been deleted. The specific amendments are as follows: the Western blot results showed that there was a good consistency between the immune response of rabbit and sheep to GTV GC protein.

2.Figure 7 & 8 are missing. Somehow we can’t find them.

Response: Thanks for the good remind. Figure 7 & 8 have been attached to files.

3.Carefully edit the manuscript to remove errors in spelling, expression, and gramma.

Response: Thanks for the suggestion. We have removed all errors in the Manuscript. For example, "fifine" revised as "fine" in line 77, "immunedominant" revised as " immunodominant" in line 266, and so on.

Minor Issues:

1.There are many errors with spelling, expression, or English gramma. These errors need to be corrected. Please carefully edit the manuscript before resubmission. For example:

Line 66: “… which is most closely related to GTV 66 evolution”

Response: Thanks, we have rewritten “which is most closely related to GTV 66 evolution” as “At present, the virus cell receptor binding domain of SFTSV is located on glycoprotein Gc” in line 65 in the revised manuscript.

Line 77: “… are as no…”

Response: Thanks, we have rewritten this sentence in line 77 of the revised manuscript. 

Line 144: “…the GTV Gc protein was designed as fifty-seven 16-mer peptides…”

Response: Thanks, we have rewritten “The GTV Gc protein was truncated into fifty-seven 16mer-peptides using modifified overlapping peptide biosynthesis” as “The GTV Gc protein was designed into fifty-seven 16mer-peptides using modified overlapping peptide biosynthesis” in line 143 of the revised manuscript.

Line 208: 16mer-peptide or 16mer peptide, with or without hyphen? Be consistent in the text and all figure legends. Please check.

Response: Thanks, we have changed all “16/10/8mer peptide” as “16/10/8mer-peptide” in the manuscript to with hyphen. 

Line 209: SDS-PAGE gel electrophoresis: GE in PAGE represents gel electrophoresis, No need to repeat. Correct this in all figure legends. Please check.

Response: Thanks, we have deleted gel electrophoresis, and revised all the “SDS-PAGE gel electrophoresis” as “SDS-PAGE” in the manuscript.

Line 210: blue.M : You need a space between the two words (letters). There are several other places with similar spelling errors in the text or figure legends. Please check.

Response: Spaces have been added and the whole manuscript been checked and corrected. For example, " blue.M " revised as " blue. M " in line 210, " bufferedsaline " revised as " buffered saline " in line 180, and so on. 

Line 212: Western blot (212), Western blotting (213, 214), western blot analysis (221), or western blotting analysis (269)? Be consistent in the text and all figure legends. Please check.

Response: Thanks, all Western blot (212), Western blotting (213, 214), western blot analysis (221), or western blotting analysis (269) in the manuscript has been unified as “Western blot analysis”. 

Line 209: What is r-clone?

Response: Thanks for the good reminding, “r-clone” revised as “recombinant clone” in lines 207, 234, and 258 in the revised manuscript.

Line 253: EGc11 and EGc12 should not have the same aa numbers from 230 to 237.

Response: Thanks, “EGc11 (230QAGMGVVG237), and EGc12 (230RSHDSQGKIS237)” revised as “EGc11 (230QAGMGVVG237), and EGc12 (303RSHDSQGKIS312)” in line 251 in the revised manuscript.

Line 259: expressed10mer-peptides

Response: Thanks, “expressed10mer peptides” revised as “expressed 10mer-peptides” with hyphen on line 257 in the revised manuscript.

Line 259: “It indicates each short peptide for numbers 10mer…” Correct expression?

Response: Thanks, we have rewritten “SDS-PAGE analysis of expressed 10mer peptides. It indicates each 10mer peptide for numbesr P3, P8, P10, P29 and P38” as “SDS-PAGE analysis of expressed 10mer-peptides for P3, P8, P10, P29 and P38” in line 257 of the revised manuscript.

2. Page numbers must be inserted on each page for submitted manuscripts.

Response: We have been inserted on each page for submitted manuscripts.

---

## [Editor Report · Decision Letter 2]

19 May 2022

PONE-D-21-02919R2Fine mapping of the antigenic epitopes of the Gc protein of Guertu virusPLOS ONE

Dear Dr. Sun,

Thank you for submitting your manuscript to PLOS ONE. After careful consideration, we feel that it has merit but does not fully meet PLOS ONE’s publication criteria as it currently stands. Therefore, we invite you to submit a revised version of the manuscript that addresses the points raised during the review process. Your manuscript and responses to previous comments have been critically assessed by academic editor.  Please include the following items when submitting your revised manuscript:A rebuttal letter that responds to each point raised by the academic editor and reviewer(s). You should upload this letter as a separate file labeled 'Response to Reviewers'.A marked-up copy of your manuscript that highlights changes made to the original version. You should upload this as a separate file labeled 'Revised Manuscript with Track Changes'.An unmarked version of your revised paper without tracked changes. You should upload this as a separate file labeled 'Manuscript'.

Specific comments:

Major comments

In Discussion, authors should clearly describe the limitations of presented study, which identified only linear epitopes of Gc proteins. It should include (1) current study does not predict B cell epitopes affected by different MHC haplotypes, (2) the mechanism of virus neutralization via Gc proteins remains unknown, and (3) the neutralizing role via identified linear epitopes of Gc proteins has not been characterized.

Minor comments

Line 21 to 22: Gc proteins are shielded by Gn protein in virions. Gn protein is responsible to receptor binding, while Gc protein serves as a fusion protein upon entry with low pH in endosomes. Although Gc proteins encode neutralizing epitope, authors should describe the role of Gc accurately with appropriate citations.

Line 47: “tick bite-transmitted” should be “tick-borne”.

Line 49: “Research studies on..” should be “Studies on..”

Line 55: “The researchers” should be “Researchers”

Line 60 – 61: Gn and Gc proteins are considered to be cleaved co-translationally, but not post-translationally.

Line 65: “At present, the virus cell receptor binding domain of SFTSV is located on glycoprotein Gc [11].” This description is not supported by cited reference.

Line 78 – 98: This paragraph describes the method for this study, which is a standard strategy to test overlapped peptides as fusion recombinants with GST. Therefore, this paragraph should be removed from introduction section. Line 100: “using the improved BSP strategy and rabbit anti-GTV-Gc pAbs” should be written as “using biosynthetic peptide (BSP) methods (Ref) and rabbit anti-GTV-Gc pAbs”.

Line 80: “The BSP method developed by Xu et al. in 2009 was created for the…” should be “The BSP method developed by Xu et al. in 2009 was used for the…”

Line 81 – 82: “the research group led by Xu Wanxiang successfully improved upon their method to conduct fine mapping..” should be “the same research group successfully mapped linear…”

Line 111 – 113: “The animal serum samples were collected using a random sampling method that did not involve sacrifice of the animals [15].” This description should clarify the following point: “serum collection was performed according to this approved animal protocol.”

Line 114: “Vectors, strains and antibodies” section requires to be separated: “Plasmids”, and “Antibodies”. PDVF membrane or ECL explanations should be moved to western blot section. “Plasmids” section should combine “Construction of the truncated Gc fragments for recombinant plasmid expression (line 129 – 137).

Line 116: “GenBank accession no. ALQ33264.1” is for protein sequence. It should be GenBank accession no. KT328592.1 (same with line 131).

Line 119: Authors should define “Gc1 and Gc2” at their first occurrence in text.

Line 120 – 121: ” Taq DNA polymerase, DNA marker, and T4 ligase (TaKaRa Co., Ltd, Dalian, China)”. This partial sentence should be corrected.

Line 120: “License number” may be removed from the text.

Line 152 – 167: “Construction” of plasmid should be combined to “Plasmid” section.

Line 166: “Heat induction at 45°C was then conducted for another 4 h.” This description requires a citation to justify the protein induction via pET plasmid in BL21 by changing incubation temperature to 45°C, rather than using standard induction using IPTG.

Line 173 – 174: “The strains expressing the correct protein size were provided to Shanghai Sangon for sequencing and strain preservation.” Authors should clarify the meaning of sentence: e.g., strains? Strain preservation?

Line 216 and 243: “To identify the minimal antigenic epitopes” should be written as “To identify the minimal linear antigenic epitopes”.

Line 237: Although text indicated that this analysis shows 8-mer peptides, whereas the legend explains that B, D, F, and H are 16-mer peptides. Authors should provide further clarification.

Line 313 – 315: “In future, serological….” This sentence should be removed from Result section.

Line 340: “Bunyaviruses are the largest family of RNA viruses…” should be corrected: e.g., “family Bunyaviridae” no longer exists.

Line 350 – 351: “could cause murine viremia, organopathological changes, and murine lesions,…” should be “could cause viremia or organopathological changes,…”

Line 396: Use of monoclonal antibody panels for SFTSV and GTV will likely map the distinct epitopes.

We look forward to receiving your revised manuscript.

Kind regards,

Tetsuro Ikegami

Academic Editor

PLOS ONE
---

## [Author Response · Author response to Decision Letter 2]

20 Jun 2022

1.In Discussion, authors should clearly describe the limitations of presented study, which identified only linear epitopes of Gc proteins. It should include (1) current study does not predict B cell epitopes affected by different MHC haplotypes, (2) the mechanism of virus neutralization via Gc proteins remains unknown, and (3) the neutralizing role via identified linear epitopes of Gc proteins has not been characterized.

Response: Thanks for the good remind. In the Discussion section we describe in detail the limitations of this study, and described in lines 394-397 as “Among study limitations, this study only identified linear epitopes of the Gc protein. Also, there are currently no B cell epitopes predicted to be affected by different MHC haplotypes. The mechanism of virus neutralization by anti-Gc protein antibodies remains unknown, and neutralization through identified linear epitopes of Gc protein has not been characterized.” 

2.Line 21 to 22: Gc proteins are shielded by Gn protein in virions. Gn protein is responsible to receptor binding, while Gc protein serves as a fusion protein upon entry with low pH in endosomes. Although Gc proteins encode neutralizing epitope, authors should describe the role of Gc accurately with appropriate citations.

Response: Thanks for the good remind. We have added corresponding references in lines 21-25.

3.Line 47: “tick bite-transmitted” should be “tick-borne”.

Response: Thanks for the suggestion. We have revised “tick bite-transmitted” as "tick-borne" in line 47.

4.Line 49: “Research studies on..” should be “Studies on..”

Response: Thanks, we have rewritten “Research studies on” as “Studies on” in line 49 in the revised manuscript.

5.Line 55: “The researchers” should be “Researchers”

Response: Thanks, we have rewritten “The researchers” as “Researchers” in line 55 in the revised manuscript. 

6. Line 60–61: Gn and Gc proteins are considered to be cleaved co-translationally, but not post-translationally.

Response: Thanks, we have revised “M segment that encodes a glycoprotein precursor that is cleaved into two mature envelope proteins (Gn and Gc) during  post-translationally modification” as “M segment that encodes a glycoprotein precursor that is cleaved into two mature envelope proteins (Gn and Gc) during co-translational modification” in line 60–61 of the revised manuscript.

7. Line 65: “At present, the virus cell receptor binding domain of SFTSV is located on glycoprotein Gc [11].” This description is not supported by cited reference.

Response: Thanks, we have added other related referencs in the revised manuscript. 

8.Line 78 – 98: This paragraph describes the method for this study, which is a standard strategy to test overlapped peptides as fusion recombinants with GST. Therefore, this paragraph should be removed from introduction section. Line 100: “using the improved BSP strategy and rabbit anti-GTV-Gc pAbs” should be written as “using biosynthetic peptide (BSP) methods (Ref) and rabbit anti-GTV-Gc pAbs”. 

Response: Thanks for the good suggestion, we have removed the description “The BSP method had the following features: (1) By selecting the truncated GST188 protein as the fusion tag for the recombinant expression of short peptides, the fragment size of the expressed fusion peptide was maintained within 21.5-22.5 kDa, which allowed its easy identification by sodium dodecyl sulfate–polyacrylamide gel electrophoresis (SDS-PAGE) and subsequent western blot analysis. This also facilitated the easy distinction of the short peptides from the proteins expressed by the host (Escherichia coli); (2) The antigenic proteins were first divided into 16mer-peptides with eight overlapping amino acid residues, following which basic and complete screening was performed for those containing BCEs; (3) Further truncation of the positive 16mer-peptides into 8mer-peptides with seven overlapping amino acid residues allowed the precise determination of the minimal epitope motif for each linear BCE [15].”, and put it in the methods section under “Strategy for designing GTV Gc overlapping peptides”. And “using the improved BSP strategy and rabbit anti-GTV-Gc pAbs” rewritten as “using biosynthetic peptide (BSP) methods and rabbit anti-GTV-Gc pAbs” in lines 80–90 in the manuscript.

9.Line 80: “The BSP method developed by Xu et al. in 2009 was created for the…” should be “The BSP method developed by Xu et al. in 2009 was used for the…”.

Response: Tanks, we revised as " The BSP method developed by Xu et al. in 2009 was created for the…” as “The BSP method developed by Xu et al. in 2009 was used for the” in line 80. 

10.Line 81–82: “the research group led by Xu Wanxiang successfully improved upon their method to conduct fine mapping..” should be “the same research group successfully mapped linear…”.

Response: Thanks, we revised “the research group led by Xu Wanxiang successfully improved upon their method to conduct fine mapping..” as “the same research group successfully mapped linear…”in line 81-82. 

11.Line 111 – 113: “The animal serum samples were collected using a random sampling method that did not involve sacrifice of the animals [15].” This description should clarify the following point: “serum collection was performed according to this approved animal protocol.”

Response: Thanks for the good reminding, we added “serum collection was performed according to this approved animal protocol.” in “Ethical statement” section in line 103 in the revised manuscript.

12.Line 114: “Vectors, strains and antibodies” section requires to be separated: “Plasmids”, and “Antibodies”. PDVF membrane or ECL explanations should be moved to western blot section. “Plasmids” section should combine “Construction of the truncated Gc fragments for recombinant plasmid expression (line 129 – 137).

Response: Thanks, the “Vectors, strains and antibodies” section was separated as “Antibodies” and other method section, “Plasmids” section was combined with “Construction of the truncated Gc fragments for recombinant plasmid expression” (line 113-116) in the revised manuscript.

13.Line 259: Line 116: “GenBank accession no. ALQ33264.1” is for protein sequence. It should be GenBank accession no. KT328592.1 (same with line 131).

Response: Thanks, we revised the GenBank accession no as “KT328592.1” line 114 in the revised manuscript.

14.Line 119: Authors should define “Gc1 and Gc2” at their first occurrence in text.

Response: Thanks, we have defined “Gc1 and Gc2” as “Gc1 (aa 1-291) and Gc2 (aa 252-549) ” in line 105 of the revised manuscript.

15. Line 120-121: “Taq DNA polymerase, DNA marker, and T4 ligase (TaKaRa Co., Ltd, Dalian, China)”. This partial sentence should be corrected.

Response: We have corrected the description in the methods section in manuscripts.

16. Line 120: “License number” may be removed from the text.

Response: We have removed from the manuscripts.

17. Line 152-167: “Construction” of plasmid should be combined to “Plasmid” section.

Response: We have constructed “Plasmid” with “Construction of the truncated Gc fragments for recombinant plasmid expression” in lines 113-116 in revised manuscript.

18. Line 166: “Heat induction at 45°C was then conducted for another 4 h.” This description requires a citation to justify the protein induction via pET plasmid in BL21 by changing incubation temperature to 45℃, rather than using standard induction using IPTG.

Response: We have revised the induction temperature “45℃” as “42℃” in lines 160 and 194 in revised manuscript. However, thermal induction was for pXXGST-3 plasmid, not pET plasmid.We have added the corresponding reference in line 161.

19. Line 173-174:“The strains expressing the correct protein size were provided to Shanghai Sangon for sequencing and strain preservation.” Authors should clarify the meaning of sentence: e.g., strains? Strain preservation?

Response: We have revised “The strains expressing the correct protein size were provided to Shanghai Sangon for sequencing and strain preservation” as “The plasmids expressing the correct protein size were send to Shanghai Sangon for sequencing” in lines 167-168 in revised manuscript.

20. Line 216 and 243: “To identify the minimal antigenic epitopes” should be written as “To identify the minimal linear antigenic epitopes”.

Response: We have revised “To identify the minimal antigenic epitopes” as “To identify the minimal linear antigenic epitopes” in line 211 in revised manuscript.

21. Line 237: Although text indicated that this analysis shows 8-mer peptides, whereas the legend explains that B, D, F, and H are 16-mer peptides. Authors should provide further clarification.

Response: Thanks for the good remind. We have corrected “16-mer peptides” as “8-mer peptides” in line 232 in revised manuscript.

22. Line 313-315: “In future, serological….” This sentence should be removed from Result section.

Response: Thanks for the good remind. We have removed the prescription “In the future, serological validation of GTV should be conducted to investigate whether these epitopes are specific to this virus and whether they can be used to detect SFTSV and GTV antibodies or to design broad-spectrum vaccines.” to lines 394-397 in revised manuscript.

23. Line 340: “Bunyaviruses are the largest family of RNA viruses…” should be corrected: e.g., “family Bunyaviridae” no longer exists..

Response: Thanks for the good remind. We have revised “Bunyaviruses are the largest family of RNA viruses” as “Bunyavirales are the largest order of RNA viruses” in line 333 in revised manuscript.

24. Line 350-351: “could cause murine viremia, organopathological changes, and murine lesions,…” should be “could cause viremia or organopathological changes,…”.

Response: Thanks for the good remind. We have revised “could cause murine viremia, organopathological changes, and murine lesions” as “could cause viremia or organopathological changes” in lines 343-344 in revised manuscript.

25. Line 396: Use of monoclonal antibody panels for SFTSV and GTV will likely map the distinct epitopes.

Response: Thanks for the good remind. We have revised the conscription in line 396 according to the comments in revised manuscript.

---

## [Editor Report · Decision Letter 3]

21 Jun 2022

PONE-D-21-02919R3Fine mapping of the antigenic epitopes of the Gc protein of Guertu virusPLOS ONE

Dear Dr. Sun,

Thank you for submitting your manuscript to PLOS ONE. After careful consideration, we feel that it has merit but does not fully meet PLOS ONE’s publication criteria as it currently stands. Therefore, we invite you to submit a revised version of the manuscript that addresses the points raised during the review process.

Specific points:

Line 22 (Abstract), and line 65 (text): Ref. 10 and 12: No experiments showed that SFTS Gc itself can bind to receptor molecules. Ref 11: Although N-glycosylation sites in Gc protein were described, the functional roles in receptor-mediated entry are speculative. Thus, there is no clear evidence to describe that “virus cell receptor binding domain of SFTSV is located on glycoprotein Gc”.  If authors just describe “N-glycosylation sites is located on SFTSV Gc glycoproteins”, and their hypothetical roles is argued, it might be fine. Glycosylation of proteins plays a role in protein folding and Golgi trafficking, which also supports virus maturation/assembly process.

Line 81-82: “More recently, the same research group successfully mapped linear of the linear BCEs of three structural proteins…” needs to be corrected.

Line 349: Authors should also mention viral fusion, which is a major role of Gc proteins.

Others: It should be clarified whether N-glycosylation sites of SFSTV Gn and Gc have been demonstrated (requiring citations) or still presumptive. If it has not been shown, all description should be written as “presumable N-glycosylation sites”.

We look forward to receiving your revised manuscript.

Kind regards,

Tetsuro Ikegami

Academic Editor

PLOS ONE
---

## [Author Response · Author response to Decision Letter 3]

4 Jul 2022

1.Line 22 (Abstract), and line 65 (text): Ref. 10 and 12: No experiments showed that SFTS Gc itself can bind to receptor molecules. Ref 11: Although N-glycosylation sites in Gc protein were described, the functional roles in receptor-mediated entry are speculative. Thus, there is no clear evidence to describe that “virus cell receptor binding domain of SFTSV is located on glycoprotein Gc”.  If authors just describe “N-glycosylation sites is located on SFTSV Gc glycoproteins”, and their hypothetical roles is argued, it might be fine. Glycosylation of proteins plays a role in protein folding and Golgi trafficking, which also supports virus maturation/assembly process.

Response: Thanks for the good remind. Indeed, there is no report proving that there is SFTSV cell receptor binding region on GC. According to your suggestion, we have revised “The viral glycoprotein Gc” as "The viral glycoprotein (GP) " in Line 22 (Abstract), and revised “the virus cell receptor binding domain of SFTSV is located on glycoprotein Gc” as “presumable N-glycosylation sites is located on SFTSV Gc glycoproteins” in line 65 (text).

2.Line 81-82: “More recently, the same research group successfully mapped linear of the linear BCEs of three structural proteins…” needs to be corrected.

Response: Thanks for the good remind. We have rewritten “More recently, the same research group successfully mapped linear of the linear BCEs of three structural proteins…” as “and they successfully mapped linear of the linear BCEs of three structural proteins” in line 81.

3.Line 349: Authors should also mention viral fusion, which is a major role of Gc proteins.

Response: Thanks for the suggestion. Virtual fusion has been added to the manuscript (line 348).

4.Others: It should be clarified whether N-glycosylation sites of SFSTV Gn and Gc have been demonstrated (requiring citations) or still presumptive. If it has not been shown, all description should be written as “presumable N-glycosylation sites”.

Response: Thanks for the suggestion. We have rewritten as “presumable N-glycosylation sites” in line 65.

---

## [Editor Report · Decision Letter 4]

11 Jul 2022

Fine mapping of the antigenic epitopes of the Gc protein of Guertu virus

PONE-D-21-02919R4

Dear Dr. Sun,

We’re pleased to inform you that your manuscript has been judged scientifically suitable for publication and will be formally accepted for publication once it meets all outstanding technical requirements.

Kind regards,

Tetsuro Ikegami

Academic Editor

PLOS ONE

Additional Editor Comments:

A few more points are noted for authors' corrections of English sentences before acceptance.

Line 81: “linear of the linear BCE” should be corrected.

Line 348: “the envelope glycoproteins are responsible for the binding and fusion of the virus particles to cellular receptors and play a protective role in passive immunization” Fusion does not occur to cellular receptors. Therefore, this sentence should be corrected: “….the binding of the virus particles to cellular receptors and subsequent viral fusion.”  “play a protective role in passive immunization” is redundant to the “Studies have shown that viral glycoproteins are the targets of neutralizing antibodies”, and can be deleted.
---

## [Editor Report · Acceptance letter]

14 Jul 2022

PONE-D-21-02919R4 

Fine mapping of the antigenic epitopes of the Gc protein of Guertu virus 

Dear Dr. Sun:

I'm pleased to inform you that your manuscript has been deemed suitable for publication in PLOS ONE. Congratulations! Your manuscript is now with our production department. 

Kind regards, 

on behalf of

Dr. Tetsuro Ikegami 

Academic Editor

PLOS ONE